# Antimicrobial Activities of Propolis in Poloxamer Based Topical Gels

**DOI:** 10.3390/pharmaceutics13122021

**Published:** 2021-11-26

**Authors:** Seong-Hyeon An, Eunmi Ban, In-Young Chung, You-Hee Cho, Aeri Kim

**Affiliations:** College of Pharmacy, CHA University, Seongnam-si 463-400, Korea; anpharm0@gmail.com (S.-H.A.); emban@korea.com (E.B.); Jaychung202@gmail.com (I.-Y.C.); youhee@cha.ac.kr (Y.-H.C.)

**Keywords:** propolis, poloxamer 188, poloxamer 407, solubility, diffusion, micelles, anti-microbacterial activity, *Staphylococcu aureus*, *Candida albicans*

## Abstract

Propolis contains a group of compounds with various activities. However, their low solubility is a drawback for the development of pharmaceutical formulations. In this study, poloxamers as a solubilizer and gelling agent were evaluated to develop a topical antimicrobial formulation of propolis. The effects of poloxamer type and concentration on the propolis solubility, release rate, and antimicrobial activities were investigated. *Staphylococcus aureus* (*S. aureus*) and *Candida albicans* (*C. albicans*) were the representative bacteria and fungi, respectively. At 5%, poloxamer 407 (P407) and poloxamer 188 (P188) enhanced the propolis solubility by 2.86 and 2.06 folds, respectively; at 10%, they were 2.81 and 2.59 folds, respectively. The micelle size in the P188 formulation increased in the presence of propolis, whereas there was no change in the P407 formulation. Release rates of propolis decreased with the P188 concentration increase, which was attributed to viscosity increase. Both P188 and P407 formulations showed antimicrobial activity against *S. aureus* in a time-kill kinetics assay. However, only the P188 formulation reduced the cell’s numbers significantly against *C. albicans*, compared to the control. We speculate that P188 mixed micelles were more effective in releasing free active compounds to exhibit anti-microbial activity compared to the P407 micelles encapsulating the hydrophobic compounds in their cores. Propolis in P188 formulation is proposed as a potential topical antimicrobial agent based on its activity against both *S. aureus* and *C. albicans*.

## 1. Introduction

Skin infections represent a significant medical problem, with at least 10,000 deaths from microbial infections in 1 million patients [1]. *Staphylococcus aureus* (*S. aureus*) and *Candida albicans* (*C. albicans*) are the most frequently isolated opportunistic bacteria and fungi species in skin infections [2,3]. Bacterial infections are often treated with topical or oral administration of antibiotics [4]. Most types of bacterial infections respond well to drugs, except for bacteria such as methicillin-resistant *Staphylococcus aureus* (MRSA), which are resistant to common antibiotics. For fungal skin infections, topical administration using antifungal sprays or creams is available [5].

Besides these well-defined antibiotics or antifungal medicines, propolis, a natural product of complex composition, has been used to treat skin infections. Propolis has historically been widely used to alleviate various diseases, including skin infections. In recent decades, several studies have shown antimicrobial [6,7,8,9], anti-inflammatory [10], immunomodulatory [11], antioxidant [12,13], and anticancer [14,15] properties of propolis, with its high levels of phenolic acids [16] and flavonoid [17]. In particular, Grecka at al., found that some fractions of flavonoids are crucial for the antibacterial activity of propolis [18]. The main components in green propolis for antibacterial activity were identified as bacharin, drupanin, *p*-coumaric acid, and artepillin C [19,20,21].

Despite the advantages of propolis, however, its low aqueous solubility is a major drawback for formulation development and functional research. Many studies have been conducted using poloxamers to improve the solubility of propolis [22,23,24]. Poloxamers are polypropylene oxide (PEO)-polyethylene oxide (PPO)-polypropylene oxide (PEO) triple block copolymers with the thermoreversible gelling property. Poloxamers can be effective in improving the solubility of poorly soluble substances [25,26] because poloxamers have the surface-active property of transforming into self-assembled micelles at concentrations above critical micelle concentration. Poloxamer 407 (P407) and poloxamer 188 (P188) are commonly used in formulations as a carrier. P407 is known to be more effective than P188 in terms of solubility enhancement and chemosensitivity. Many researchers have focused on P407 as a solubilizer [27,28] and/or a thermoreversible gelling agent [29,30,31,32] and developed the antibacterial propolis formulations using P407 alone or in combination with P188. All these studies focused on the antibacterial property of propolis in the poloxamer formulation rather than its antifungal activity. One research project reports that propolis could not inhibit the growth of *C. albicans* [33,34].

In this study, various propolis formulations in P407 or P188 have been evaluated in terms of their propolis solubility, release rate, and in vitro growth inhibition against the representative bacteria and fungi, *S. aureus* and *C. albicans*, respectively. We have used Brazilian green propolis produced by *Apis mellifera* from *Baccharis dracunculifolia*, common in the Minas Gerais state [35], and a Korean propolis extract. This is the first study to demonstrate the importance of formulation in evaluating the antimicrobial activity of propolis and the synergy between poloxamer with propolis on antimicrobial activity, especially against *C. albicans*. 

## 2. Materials and Methods

### 2.1. Materials

Poloxamer P407 and P188 were obtained from BASF Co. (Ludwigshafen, Germany). Carbopol 971P Ultraz 10 NF was provided by Lubrizol Co. (Wickliffe, OH, USA). Green propolis extract in propylene glycol (30%, *w*/*w*) from Essenciale ltda (Minas Gerais, Brazil) was supplied by 7Tree Co. (Seongnam, Korea). Korean propolis extract in ethanol was from Korea Yangbongnh. Malt and yeast extract were purchased from NEOGEN Co. (Lansing, MI, USA). Tryptone was purchased from Duchefa Biochemie BV. (Haarlem, The Netherlands).

### 2.2. Measurement of Propolis Solubility in Various Poloxamer Solutions

Various compositions listed in Table 1 were centrifuged at 18,000× *g* for 1 h, and their supernatants were collected. The supernatants were diluted 200-fold with a solvent (1:1 = DW:MeOH), and the absorbance was measured with a microplate reader (Synergy Mx, BioTeK, Winooski, VT, USA) at 330 nm. There was no interference by polymers in absorption at the same wavelength. A calibration curve of propolis in the same solvent was constructed to determine the solubility of propolis in each formulation.

### 2.3. Preparation of Propolis-Poloxamer Formulations

Stock solutions of P188 (40%) and P407 (25%) were prepared in distilled water (DW). They were diluted to the final polymer concentrations shown in Table 2 and the green propolis solution was added to obtain the final propolis concentration of 10% (*w*/*w*). For the formulation containing Carbopol, an aliquot of 1N NaOH was added in the last step to titrate the pH to 5.5. The mixtures were then stored overnight in a cold room at 4 °C with stirring to complete hydration.

### 2.4. Measurement of Release Profile, Viscosity, and Micelle Characterization

A release test was carried out using Transwell inserts (SPLInsert™, SPL Life Sciences Co., Ltd., Pocheon-si, Korea) with a membrane cut-off of 0.4 μm as previously described [36]. 100 mg of each formulation listed in Table 2 was loaded into the inserts, and the wells were filled with 900 μL of 30% sucrose solution to minimize the effect of osmotic pressure [36]. The plates were shaken at 37 °C for 8 h while sampling 200 μL at 0.25, 0.5, 1, 2, 4, and 8 h. At each time point, the inserts were transferred to the next well containing 900 μL of fresh media. The amount of propolis released in each well was determined by the method described above for the solubility measurement. 

The viscosities of vehicle controls (VC) of various formulations without propolis listed in Table 2 were measured at 30 °C and 37 °C using a rotary viscometer (DV-II+ Pro, AMETEK Brookfield, Chandler, AZ, USA). The measurement was done at 30RRM using the S61 spindle, with the torque ranging 20~90%.

The formulations were characterized in terms of micelle size and polydispersity by dynamic light scattering, using Zetasizer Nano ZS (Malvern Instruments Ltd., Almelo, UK) with back-scatter detection at a scattering angle of 173°. The samples in polystyrene disposable cuvettes were allowed to equilibrate at 37 °C for 5 min before analysis. Measurements were carried out for P188 and 407 formulations (10%), containing 10% of propolis propylene glycol extract or ethanol extract. In addition, aqueous solutions of P188 and P407, and vehicle controls without propolis, were also characterized. Samples diluted by 10-fold and 40-fold with YM medium were also included for the characterization. All samples were centrifuged at 18,000× *g* for 10 min to avoid interference from any particulate matter. The measurements were performed on three independent parallels.

### 2.5. Measurement of Trypan Blue Diffusion Pattern

Diffusion of trypan blue was visualized on agar plates and Transwell inserts with and without agar-coating. Agar coating on Transwell inserts was prepared to mimic the diffusion in agar plates by hardening the agar solution used in the antimicrobial test to a thickness of 0.3 cm on the membrane of Transwell inserts (SPLInsert™, SPL Life Sciences Co., Ltd., Korea). Agar plates were spotted with 5μL of trypan blue-loaded F2, F6, and F2 + Carbopol formulations in Luria-Bertani (LB) and yeast malt (YM). Photographs were taken after standing for 18 h at 37 °C and 30 °C, respectively. For diffusion across the Transwell inserts, 100 μL of F2, F6, or F2 + Carbopol mixed with 900 μL of LB or YM were loaded on the inserts. Photographs were taken after 18 h for Transwell inserts with agar coating and after 2 h for Transwell inserts without agar coating.

### 2.6. Bacterial Strains and Culture Conditions

The bacterial strain *Staphylococcus aureus Newman* (*S. aureus*) and fungal strain *Candida albicans ATCC 10231* (*C. albicans*) were used in this study. *S. aureus* and *C. albicans* were grown at 37 °C and 30 °C in Luria-Bertani (LB) (1% tryptone, 0.5% yeast extract, and 1% NaCl) broth and yeast malt (YM) broth, respectively. Overnight-grown cultures were used as inoculum (1.6 × 10^7^ CFU/mL) into fresh LB or YM broth and grown at 37 °C or 30 °C in a shaking incubator until the logarithmic (OD_600_ = 1.0) phase, and then the cell cultures were used for the experiments described herein.

### 2.7. Antimicrobial Susceptibility Test

For the antimicrobial susceptibility test, cell lawns were generated by overlaying LB or YM agar plates with soft LB agar (0.7%) containing about 1 × 10^8^ cells. After air-drying the plate for 1 h, aliquots (5 μL, equivalent to 500 μg propolis) of various formulations listed in Table 2, VC without propolis, and unformulated propolis dissolved in 10% DMSO were dotted without a filter disk placed on a cell plate. After that, the plate was incubated in an incubator at 37 °C or 30 °C for 18 h (Figure 1). The diameter of the inhibition zone (clear zone around each well) was measured with a caliper in millimeters.

### 2.8. Antimicrobial Kinetics Assay

For antimicrobial kinetics assay, cells were cultured to OD_600_ of 1.0 in LB or YM broth and diluted to approximately 1 × 10^6^ CFU/mL suspension. Various formulations and VC without propolis were diluted in the broth. They were then mixed with the cell suspension at a ratio of 1:1. Each mixture (100 μL) was inoculated into a 96-well microtiter plate. The final concentration of propolis in each well was 2500 μg/mL (Figure 1). Unformulated propolis propylene glycol extract was dissolved in 10% DMSO. A separate experiment confirmed that the DMSO at this concentration did not affect the cells. Inoculated 96-well plates were incubated for 18 h at 37 °C for *S. aureus* and 30 °C for *C. albicans*, while measuring their OD_600_ every 20-min using an Epoch™ 2 Microplate Spectrophotometer (BioTek Inc.). 

### 2.9. Statistics

Statistical analysis was performed using GraphPad Prism software-8 (GraphPad Prism Software, Inc., San Diego, CA, USA), and data are presented as mean ± SD. The comparisons between two groups were determined by the two-tailed student’s *t*-test. For comparisons among more than two groups, one-way ANOVA tests were performed followed by Dunn’s multiple comparison tests as a post hoc test. The comparisons were considered statistically significant when the *p*-value was less than 0.05 (*p* < 0.05).

## 3. Results

### 3.1. Effect of Poloxamer Type and Concentration on Propolis Solubility

A propolis propylene glycol extract (30%, *w*/*w*) provided by the supplier was added in the aqueous solutions of P188 and P407 to make the final concentrations of poloxamers ranging from 5% to 30% and from 5% to 10%, respectively (Table 1). Both P188 and P407 had a positive effect on the solubility of propolis (Figure 1). At 5% and 10%, P407 was more effective than P188 in solubility enhancement. These results are consistent with our previous results [28]. The maximum solubility of propolis was obtained with 5% P407 and 10% P188. Samples with more than 10% of P407 or 30% of P188 were excluded from the measurement because they appeared as a clear gel. P407 solution at a specific concentration turns to a hydrogel above a certain temperature, i.e., the gelation temperature. To evaluate the gelling behavior of 20% P407 solutions containing 6%, 10%, or 20% propolis extraction, their fluidity was compared at room temperature (RT) and 37 °C (Appendix A). The 20% P407 solution remained as a solution at RT and changed to a gel at 37 °C. In addition, a 20% P407 solution containing 6% propolis extract was changed to a gel at both room temperature and 37 °C. However, such thermoreversible behavior of 20% P407 solution was not observed at 10% or 20% propolis concentration and showed high rigidity at RT and 37 °C because of the increased content of propylene glycol. More detailed results on the rheological properties of poloxamer solutions, or gels containing various concentrations of propolis extracts, are described in Appendix A.

### 3.2. Effect of Poloxamer Type and Concentration on Release of Propolis

The release rates of propolis from poloxamer formulations appeared to decrease with increasing poloxamer concentration for both P188 and P407 (Figure 2A,B). There was a statistically significant correlation (R2 = 0.9627, *p* = 0.0188) between P188 concentration and the release rates of propolis (Figure 2A Insert). There was no statistically significant difference in the release rate between F1 and F5 or between F2 and F6. 

The viscosity of vehicle controls increased with the increasing concentration of poloxamer, and the viscosity of P407 was higher than that of P188 at the same concentration (Figure 3). The viscosity of VC-F2 in the presence of Carbopol was above the maximum measurable value in the experimental setting.

The Z-average micelle size and PDI of various solutions and formulations are presented in Table 3. Effects of various parameters such as poloxamer type, the presence of propylene glycol and ethanol, propolis, and the dilution factor were evaluated. The size distribution profiles for representative samples are presented in Appendix A. The micelle size is larger in P407 than in P188 formulations. The size distribution profiles of P407 solutions or formulations were unimodal with the size range of 24–30 nm, whereas those of P188 was bimodal. And most P188 micelles were in a range of 5–8 nm, but some fractions of samples were larger than 100 nm. In addition to such difference in size distribution profiles, another striking difference between P407 and P188 formulations was the changes in micelle size depending on the presence of propolis. While undiluted samples or 40-fold diluted samples were not good for DLS measurement because of optical interference or precipitation, the 10-fold diluted samples allowed the comparison between P407 and P188 in terms of propolis dependency of micelle size. P188 micelle size increased by 7 times in the presence of propolis; however, P407 did not show a significant increase in the micelle size.

### 3.3. Diffusion Patterns of Trypan Blue in P188, P407, and P188 with Added Carbopol Formulation

We compared the diffusion pattern of trypan blue dissolved in P188, P407, or P188 containing Carbopol formulations. As shown in Figure 4, the diffusion of trypan blue in 10% P188 was faster than that in 10% P407 in both agar plate and broth conditions. 10% P188 containing Carbopol showed the slowest diffusion.

### 3.4. Antimicrobial Susceptibility Test of Propolis-Poloxamer Formulations

The antimicrobial activities of propolis in various formulations of P188 and P407 (Table 2) were evaluated against representative bacteria and fungi, *S. aureus* and *C. albicans*, respectively. *S. aureus* was more sensitive to propolis than *C. albicans* with a larger zone of inhibition (Figure 5). All propolis formulations in P188 inhibited the growth of *S. aureus* with the best activity in 10% P188, whereas the inhibition zone of growth was significantly reduced in 10% P407 (Figure 5A,C). The inhibition zone of *C. albicans* growth was smaller in 20% P188, and there was no growth inhibition in 30% P188 or both 5% and 10% P407 formulations (Figure 5B,D). Antimicrobial test data of vehicle controls without propolis or propolis without poloxamers are described in Appendix A. When P188 or P407 solution at 5% and 10% without propolis were placed directly onto the surface of *S. aureus* and *C. albicans*-containing agar plate, the inhibition zone was not detectable. In addition, when propolis without P188 or 407 was placed onto the surface of *C. albicans* and *S. aureus*-containing agar plate, the inhibition zone was observed for both *C. albicans* and *S. aureus*. Their inhibition zones were smaller than those of P188 formulations (F1, F2), and larger or clearer than those of the P407 formulations (F5, F6).

### 3.5. Time-Kill Kinetics Assay of Propolis-Poloxamer Formulations

The antimicrobial activities of propolis incorporated in various poloxamer formulations were compared through the time-kill kinetics assay (Figure 6 and Figure 7). The control groups were the *S. aureus* and *C. albicans* suspended in broths without propolis or poloxamers. All P188 formulations showed better antibacterial activity than propolis dissolved in DMSO (Figure 6A). Propolis in 5% to 20% P188 completely inhibited bacterial growth for 18 h. On the other hand, their antifungal activity against *C. albicans* was observed up to 12 h, and the activity of propolis in 20% or 30% P188 was slightly better or worse than that of DMSO (Figure 6B). In contrast to the propolis formulation in P188, those in P407 showed worse antibacterial and antifungal effects than propolis dissolved in DMSO (Figure 6C,D). DMSO at this concentration did not affect the cells. In addition, vehicle controls of 5% and 10% P188 showed statistically significant inhibition *against C. albicans* compared to the control group (Appendix A). A time-kill kinetics assay in the presence of Carbopol as a gelling agent showed that the antibacterial activity of 10% P188 formulation was not affected, but the antifungal activity was negatively affected by Carbopol (Figure 7).

In order to evaluate the effect of poloxamers on another source of propolis, the time-kill kinetics of propolis extracted with ethanol were also performed in P188 and P407 solutions. Experimental details are described in Appendix A. For this experiment, propolis concentration in each well was lowered to 625 μg/mL for meaningful comparison between propylene glycol- and ethanol extract. Even at this low concentration, both propylene glycol and ethanol-extracted propolis showed complete inhibition of *S. aureus* when formulated in P188 at 5% or 10% (F1-PG, F1-EtOH, F2-PG, and F2-EtOH in Figure 8A,C). P407 formulations of propylene glycol-extracted propolis (F5-PG and F6-PG) were not as good as P188 formulations but showed better activity than DMSO solution (Figure 8A), while those of ethanol-extracted propolis showed complete inhibition (Figure 8C). All these formulations showed better activity against *C. albicans* compared to DMSO solutions and P188 was more effective than P407 in improving the activity of both propylene glycol and ethanol-extracted propolis.

## 4. Discussion

Several studies in the literature describe a wide range of antimicrobial activities of propolis in various formulations [6,7,17,33,37,38]. Previously, we explored P188 and P407 to effectively enhance the solubility of emodin, a hydrophobic substance derived from plants [28]. In the present study, propolis extract in propylene glycol was formulated in P188 or P407 to enhance the solubility of active compounds in propolis, and thereby to develop it as a topical antimicrobial therapeutic.

Z-average micelle sizes in 10% P407 and P188 solutions determined in the present study, 6nm and 25 nm, respectively, agree with the literature values [39,40]. Larger micelles can encapsulate more hydrophobic compounds in the micelle cores. Therefore, higher solubility of propolis in P407 can be attributed to the micelles large enough to incorporate the active compounds of propolis without an increase in the micelle size. In contrast, the increase of P188 micelle size in the presence of propolis suggests formation of mixed micelles composed of P188 and various hydrophobic compounds in propolis. Singla et al. also reported that incorporation of hydrophobic compounds such as lamotrigine in P188 increased the micelle size of P188 [39].

Fick’s law of diffusion states that the driving force of molecular diffusion is the concentration gradient across the two regions. The flux is related to the diffusivity of diffusing molecules in the media [41]: J = D·dC/dx, where D is the diffusivity of the diffusing molecules, and dC/dx is the concentration gradient. Diffusivity is inversely proportional to the viscosity of diffusion media, according to Wilke and Chang [41]. A good correlation between the propolis release and the P188 content in the formulations (Figure 2A, Insert) can be attributed to the higher viscosity in the formulations of higher P188 content. However, the propolis release rates between P188 and P407 of the same concentration (F1 vs. F5, or F2 vs. F6) did not differ significantly, despite the differences in their viscosities. If the release rates were determined by the viscosity only, F5 or F6 would exhibit lower release rates than F1 or F6, respectively. Instead, their release rates were not different, because the higher solubility of propolis in F5 or F6 mitigated the negative effect of higher viscosity on their release rates. In contrast, trypan blue in the agar plates and across the Transwell inserts alike show slower diffusivity in P407 than in P188. Unlike propolis, trypan blue is water-soluble and therefore the media viscosity is the only determinator of diffusion rate in this case. These results demonstrate the impact of water solubility of diffusing molecules on their release rates relative to the media viscosity.

The previous study on the effect of release media during the release test of poloxamer gel emphasized the importance of the judicious selection of release media for topical formulations depending on the skin condition [36]. In the present study, 30% sucrose solution was used for the release test using Transwell inserts. Otherwise, the osmotic pressure difference between the dosing formulation and the release media could overestimate the release rate.

The antimicrobial activities of propolis formulations in P188 and P407 were tested by two methods: the measurement of growth inhibition zone in agar plates and the time-kill assay performed in cell suspensions. The first method using the agar plates is appropriate to understand the effect of formulation viscosity on their activities. Better antibacterial activity of P188 formulations compared to P407 in this setting (Figure 5) can be attributed to the faster release and diffusion of propolis through the agar plate in P188. On the other hand, the time-kill assay in the cell suspension would mainly be affected by the solubility of propolis when the system is diluted enough to minimize the viscosity. The activity of the P407 formulation was even worse than the DMSO solution formulation, which became turbid when diluted with the broth because of the low aqueous solubility of propolis. Propolis solubility in the formulation alone cannot explain such negative activity of P407 formulations in the time-kill assay. One plausible explanation for the low activity of P407 formulations despite the high solubility of propolis in p407 would be that the level of free active compounds available for anti-microbial activities was low due to the effective incorporation of propolis in P407 micelles. Solubility of propolis in poloxamers is governed by micelle formation as discussed above. One can speculate that effective encapsulation of propolis in P407 micelle cores would result in slow dynamic equilibrium between free active compounds in solution and those in micelle cores. In contrast, mixed micelles of P188 with the active compounds of propolis would release those active compounds faster if they were not entirely incorporated in the micelle cores. It would be an interesting future research subject to investigate the effects of micelle structures on the equilibrium dynamics between the hydrophobic compounds in the micelle cores and those in solution.

There was a difference in the activity between F2 with and without Carbopol (Figure 7). Although one might neglect the effect of formulation viscosity in the time-kill assay when the formulations were sufficiently diluted, Carbopol might have negatively affected the activity of F2 by increasing the viscosity of the medium, because Carbopol increased the viscosity of F2 more than thirty folds. This result indicates that formulation viscosity can affect the time-kill assay results when the viscosity difference is significant enough.

The difference in antimicrobial activity against a particular strain type appears to be related to the minimal inhibitory concentration (MIC) of each strain, and MIC of *S. aureus* was four-fold lower than that of *C. albicans* [33]. The efficacy of some antimicrobial agents can be correlated, not to the high peak concentration, but to the time to keep the concentration of the antimicrobial agent above the MIC [42]. The growth inhibition zone test and the time-kill assay results indicate that the P188 formulation delivered propolis above MIC values of both *S. aureus* and *C. albicans*, whereas the P407 formulation delivered propolis only above MIC of *S. aureus.* Notably, the positive activity of the P188 formulation against *C. albicans* was not only due to propolis, but also P188 as vehicle per se, as demonstrated in Figure 6B and as described in Appendix A. To the best of our knowledge, the present study is the first to present a propolis formulation in P188 with antifungal activity against *C. albicans.* The antifungal activities of propolis reported in the literature vary widely due to the differences in the source of propolis and the compositions of propolis and experimental settings. Dantas Silva, et al. and Popova et al. did not observe any inhibitory activity of Brazilian propolis and Mediterranean propolis against *C. albicans*, respectively [33,43]. However, there are other reports on the antifungal activity of propolis extracts against *C. albicans* [44,45]. In the present study, not only the Brazilian propylene glycol extract but also the Korean ethanol-extract propolis showed similar trends in terms of higher activity against *S. aureus* than *C. albicans*. The results of our study agree with published data that indicated high antibacterial activity of propolis against *S. aureus* and low antifungal activity of propolis against *C. albicans* [46,47]. In addition, P188 was a better solubilizer than P407 for both different sources of propolis tested in this study to develop antimicrobial topical formulations.

Taken together, the faster release rate of propolis from P188 formulation resulted in better antimicrobial activity compared to P407 formulation, which had a higher viscosity than P188. Furthermore, the micelle structures appeared to affect the activity of propolis formulated in poloxamers. The study results suggest that solubility enhancement alone cannot achieve an optimized formulation of poorly soluble compounds. Both poloxamer type and concentration were essential parameters to develop an effective antimicrobial formulation of propolis with antimicrobial activities against both bacteria and fungi. Further studies before clinical translation of the present results would include cytotoxicity tests in human cells and skin permeation tests. In addition, activity against other pathogenic microorganisms is warranted to propose propolis in P188 formulation as a potential antimicrobial agent.

## 5. Conclusions

Both poloxamer type and its concentration were essential parameters to develop an effective antimicrobial formulation of propolis with antimicrobial activities against both anti-bacteria and anti-fungi. Propolis in P188 formulation is proposed as a potential topical antimicrobial agent based on its activity against both *S. aureus* and *C. albicans*.

## Data Availability

All data available are reported in the article.

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
