# Peer review of "Antimicrobial Activities of Propolis in Poloxamer Based Topical Gels"

_pharmaceutics, 2021, doi:10.3390/pharmaceutics13122021_

Round 1

Reviewer 1 Report

In this manuscript, An, Ban and Kim are reporting the preparation, characterization and antimicrobial activity of poloxamer-based gels containing propolis. The methodology is in accordance with the questions raised by the authors. However, authors must address the following issues before publication:  

1 - Poloxamers are well-known materials that can self assemble into micelles above a CMC. As far as the reviewer is concerned, the concentration used is above CMC. It would enrich the manuscript if the authors characterize the micelles, by measuring zeta-average diameter, poly-dispersity index, surface charge and others.

2- Authors reported that some of their formulation were excluded (Results, first paragraph) because they appeared as clear gel. Poloxamer formulations are also well-known to have a gelification point, according to the concentration and temperature. Authors should measure the gelification point in which the concentration and the temperature of the poloxamer suspension would turn into gel. This is of most interest in the formulation of gels. Methods are easily found elsewhere.

3- Topic 2.7: How did the authors measured the zone of inhibition? Software? Caliper? This should be stated in this section.

4– Why the authors used students T test instead of ANOVA followed by post-test? Some experiments you must compare all the treatments.

5- Please, insert the description of controls used in each experiment. For example: "VC represent vehicles of F1, F2, F5, and F6 without propolis."

6- Topic 3.2:  was the difference of release profile between poloxamers statistically significant? Please analyze if there is statistical significance (Figure 2), and insert the description in the text so the information is complete.

7– Figure 2: merge Figure A and B. Improving the description in the legend would also provide better understanding. 

8- Topic 3.4: Please, include the controls in this experiment. How the formulation without propolis perform per se? How is it the activity of propolis alone?

9- What are the active constituents of the propolis responsible for the antimicrobial activity? Authors should point the chemical constituents of propolis.

10 – The authors suggested further studies using other pathogens. Wouldn’t it be more important to test this formulation in human cells and evaluate its cytotoxicity and skin permeation so they could ensure the formulation safety?

11- Where is the statistics about P188 vehicle being significantly different and making a difference against C. albicans? (Discussion). 

Reviewer 2 Report

Review of the article: “Antimicrobial activities of propolis in poloxamer based topical gels”

Submission ID - pharmaceutics-1419308

In my opinion using of poloxamer for preparing new formulations of propolis is an interesting idea. Some of results presented in this manuscript are promising. However, I have some important critical remarks about the plan of the study:

  1. The authors used only one sample of propolis for their study.
  2. In fact the authors did not use the raw material (propolis) but they used “. Green propolis solution in propylene glycol (30%, w/w) from Essenciale ltda (Minas Gerais, Brazil)” – some important ingredients from raw material could not be extracted with propylene glycol.
  3. In my opinion activity of these new formulations should be compared to the activity of extracts produced (from raw material) with ethanol. This is the most common formulation of propolis.

Detailed comments:

Abstract

Generally the abstract is well prepared. However, some, most important results (data) should be included, not only general conclusions (e.g. However, P188 formulation with lower viscosity than P407 showed better release profiles, which was positively correlated to better antimicrobial activity.)

Introduction

Lines 36-37 “properties of propolis with its high levels of phenolic acids [11] and flavonoid [12].” – some more details about active ingredients of propolis should be presented. E.g. Grecka at al., found that some fractions of flavonoids are crucial for antibacterial activity of propolis. Other authors also discussed correlation between activity and chemical composition of this product.

Lines 50-51 – I do not understand this conclusion. There are a plenty of publications that confirm antifungal activity of propolis including Candida spp.. – some of them could be cited. Moreover, I have gone through citation 24 and activity of propolis against Candida was not investigated in this study. Please check it carefully.

Materials and methods

Line 124 – RPMI medium would be more suitable for this assay

Lines 1290130 – did the authors observe any precipitation of the propolis components from the mixture

Results

In my opinion the quality of figures 4A, 4B, 5A and 5B are not clear. On the basis of these pictures it is difficult to do verification of the data presented in the graphs, particularly 5C and 5D.

Discussion

Discussion is not well prepared. In fact between lines 211-264 the authors presented description of results not discussion.

Final decision – major revision.

Round 2

Reviewer 2 Report

The authors have addressed all my critical comments. In my opinion the revised version is importantly better than the previous one. The current version of the manuscript can be accepted.